# Cellular parameters shaping pathways of targeted protein degradation
Annabel Cardno ⓘ , Bryony Kennedy & Catherine Lindon ⓘ ✉

In recent years the development of proteolysis-targeting chimeras (PROTACs) has enhanced the field of ubiquitin signalling through advancing therapeutic targeted protein degradation (TPD) strategies and generating tools to explore the ubiquitin landscape. However, the interplay between PROTACs and their substrates, and other components of the ubiquitin proteasome system (UPS), raises fundamental questions about cellular parameters that might influence the action of PROTACs and the amenability of a given target to PROTAC-mediated degradation. In this perspective we discuss examples of cellular parameters that have been shown to influence PROTAC sensitivity and consider others likely to be important for PROTAC-mediated target degradation but not yet routinely considered in design of novel TPD strategies: Target localisation and accessibility on the one hand, and expression patterns, localisation and activity of E3 ligases, deubiquitinases (DUBs) and wider ubiquitin machinery on the other, are critical parameters in the exploitation of PROTACs, and establishing a better understanding of these parameters will facilitate the rational design of PROTACs.

The ubiquitin-proteasome system (UPS) is a key intracellular pathway responsible for the maintenance of cellular proteostasis, acting to achieve the degradation of ubiquitin-tagged protein substrates at the 26S proteasome. A ubiquitin-activating enzyme (E1) activates ubiquitin, a 76-amino acid polypeptide, by converting it to a C-terminal thioester. Activated ubiquitin is then transferred to a ubiquitin-conjugating enzyme (E2), followed by transfer onto a lysine residue of the substrate protein, a process that requires a ubiquitin ligase (E3) to bring the E2 and substrate within close proximity of one another[1]. Ubiquitin molecules can be serially added onto each other through internal lysine residues (or the N-terminal amino group) to form a polyubiquitin chain that can be either homotypic or heterotypic (where more than one lysine is modified) and therefore branched[2]. Ubiquitin contains seven lysine residues, creating an inexhaustible range of possible linkage topologies for heterotypic polyubiquitin chains which leads to a diversity of cellular outcomes (sometimes referred to as the 'ubiquitin code'). Chains linked through K48 have been most strongly associated with targeting substrates for proteasomal degradation[3], although other linkages such as K11 are also important, particularly during the cell cycle[4,5].

Over recent years, there has been an explosion in the study of how the UPS can be harnessed for the development of targeted protein degradation (TPD) tools, including proteolysis-targeting chimeras (PROTACs). PROTACs are heterobifunctional molecules with discrete binding moieties for an intended target protein substrate and for an E3, connected by a chemical linker, and thereby bring the target protein into proximity of the E3. A successful PROTAC mediates formation of a ternary complex between the two that is stable enough for ubiquitination of the target protein to occur, leading to its elimination via the proteasome[6–8] (Fig. 1).

The first successful proof of concept for TPD, more than 20 years ago, was a peptide-based PROTAC consisting of the MetAP-2 inhibitor ovalicin linked to a phosphopeptide motif of IκB, which is recognised by the SCF F-box protein β-TRCP[9]. In more recent years, the field has been transformed through discovery of small molecule ligands for cullin-RING E3 ubiquitin ligase (CRL) components such as CRBN and VHL, substrate adaptors for CRL2 and CRL4 respectively. In particular, the discovery of the mechanism of action of the immunomodulatory drugs (IMiDs) such as pomalidomide, thalidomide and lenalidomide in binding CRBN as a molecular glue, to induce ubiquitination and proteasomal degradation of zinc-finger transcription factors (IKZF1, IKZF3)[10,11] demonstrated that CRBN can be effectively redirected to degrade so-called 'neosubstrates'. These drugs subsequently provided the means to harness CRBN for PROTAC-mediated degradation of targets and sparked an expansion of research driven by the therapeutic potential of small molecule PROTACs. There are currently over 20 PROTACs in clinical trials (Table 1). Notably, ARV-110 and second-generation ARV-766 from Arvinas against the androgen receptor (AR)[12,13], and ARV-471 against the estrogen receptor (ER)[14] demonstrate potent and selective degradation of their targets in preclinical models and are now being progressed to phase III clinical trials (NCT03888612, NCT0506714, NCT05654623).

PROTACs offer many advantages over traditional small molecule inhibitors[15], allowing transient target interaction and catalytic action for

Department of Pharmacology, University of Cambridge, Cambridge, UK. ✉e-mail: acl34@cam.ac.uk

efficacy at low doses, potentially improving safety profiles. They show increased substrate selectivity[16] and can target previously 'undruggable' proteins, as demonstrated with PROTACs for STAT3 and c-Myc[17,18]. Much of PROTAC research to date has been focused on optimisation of PROTAC properties such as target engagement and specificity. The composition and length of the linker between binding moieties for substrate and E3 strongly influence the activity of TPD tools by contributing to formation of a productive ternary complex which is cooperative in nature yet flexible enough to allow ubiquitination of targets[19,20]. This process has been shown in many studies to correlate with the efficiency of neosubstrate degradation[6,21,22].

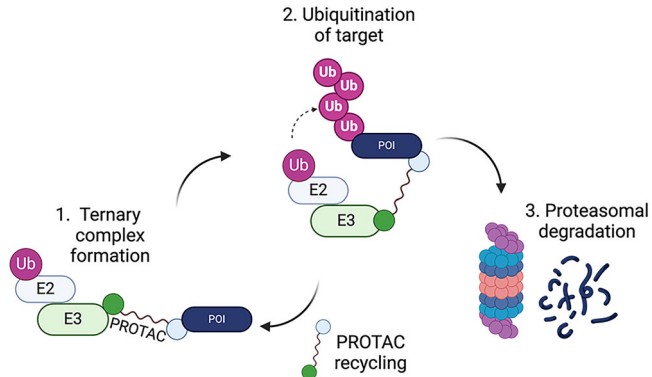

**Fig. 1 | Schematic of PROTAC activity.** The heterobifunctional small molecule promotes ternary complex formation between E3 and a target protein (1) to bring about ubiquitination (2) then degradation (3) of the target as a 'neo-substrate' of the E3. Created in BioRender. Cardno, A. (2025) https://BioRender.com/7ggcsvs.

Despite the large size of PROTACs and the associated challenges with delivery to targets in situ, many PROTACs have displayed efficient cell and tissue uptake, good activity in in vivo models and reasonable oral bioavailability[23,24]. However, while several PROTACs have entered clinical trials in recent years, there is a poor understanding of cellular parameters governing the success of PROTACs in vivo. Early-stage screening of compounds for TPD activity often measures target protein degradation as the endpoint, bypassing steps between target engagement and degradation that may include the rate-limiting step of protein degradation for a particular target in a particular cellular context. This means there is little information available to explain why compounds optimised in vitro may fail in in cellulo assays of target degradation. For the 21 PROTACs in Table 1, there are likely many more that have failed to show sufficient activity in cellular assays.

Even PROTACs that have favourable pharmacokinetics and have entered clinical trials, such as ARV-766, elude fundamental cell biology questions, with limited information available detailing the localisation of TCF and target degradation. Current anti-androgens may act to inhibit translocation of cytoplasmic AR to the nucleus as well as inhibiting transcriptional activation by the nuclear pool of AR[25]. Other ligands (and potential PROTAC warheads) similarly act to modulate cellular behaviour of their targets, for example the EGFR inhibitor AG-1478 prevents the nuclear translocation of EGFR stimulated by EGFR ligand sHB-EGF, trapping EGFR in the cytoplasm and at the plasma membrane[26]. The choice of target binder in PROTAC design may therefore influence localisation of TCF, potentially a critical parameter in determining the degradability of neosubstrates.

Many cellular factors will influence TCF and substrate ubiquitination and processing as part of the complex cellular response to PROTAC treatment. These factors may be target-specific or more universally applicable to the UPS components harnessed in TPD strategies. A growing

## Table 1 | PROTACs currently in clinical trials

| Name | Company | Target | E3 ligase | Indication | Phase |
|---|---|---|---|---|---|
| ARV-110 | Arvinas | AR | CRBN | mCRPC | II NCT03888612 |
| ARV-393 | Arvinas | BCL6 | CRBN | R/R NHL, R/R AITL | I NCT06393738 |
| ARV-471 | Arvinas | ER | CRBN | ER + /HER2- breast cancer | III NCT05654623 |
| ARV-766 | Arvinas | AR | CRBN | mCRPC | II NCT05067140 |
| ASP3082 | Astellas Pharma | KRAS$^{G12D}$ | UD | Solid Tumour | I NCT05382559 |
| BGB-16673 | BeiGENE | BTK | CRBN | B-cell malignancies | II NCT05294731 |
| CC-94676 | BMS | AR | UD | mCRPC | I NCT04428788 |
| CFT-8634 | C4 Therapeutics | BRD9 | CRBN | Synovial sarcoma, SMARCB1-null solid tumours | II NCT05355753 |
| CFT-1946 | C4 Therapeutics | BRAF | CRBN | BRAF$^{V600E}$-driven cancers | II NCT05668585 |
| CG001419 | Cullgen | NTRK | CRBN | Solid tumours | II CTR20222742 |
| DT-2216 | Dialectic Therapeutics | BCL-XL | VHL | T-cell lymphomas | I NCT04886622 |
| GT-20029 | Kintor Pharmaceutical | AR | CRBN | Acne vulgaris | I NCT05428449 |
| HP518 | Hinova | AR | UD | mCRPC | I NCT06155084 |
| HSK-29116 | Haisco | BTK | CRBN | B-cell malignancies | I NCT04861779 |
| KT-253 | Kymera Therapeutics | MDM2 | UD | R/R high grade myeloid malignancies and solid tumours | I NCT05775406 |
| KT-333 | Kymera Therapeutics | STAT3 | VHL | Liquid and solid tumours, T cell lymphomas | I NCT05225584 |
| KT-474 | Kymera Therapeutics | IRAK4 | CRBN | Hidradenitis suppurativa | II NCT06028230 |
| NX-2127 | Nurix Therapeutics | BTK, IKZF1, IKZF3 | CRBN | R/R B cell malignancies | Ia/Ib NCT04830137 |
| NX-5948 | Nurix Therapeutics | BTK | CRBN | R/R B cell malignancies | Ia/Ib NCT05131022 |
| PRT3789 | Prelude Therapeutics | SMARCA2 | UD | NSCLC | I NCT05639751 |
| RO7656594 | Roche | AR | UD | Advanced or metastatic prostate cancer | I NCT05800665 |

*UD* undisclosed, *mCRPC* Metastatic Castration-Resistant Prostate Cancer, *R/R NHL* Relapsed/Refractory Non-Hodgkin Lymphoma, *R/R AITL* Relapsed/Refractory Angioimmunoblastic T-Cell Lymphoma, *NSCLC* Non-small cell lung cancer.
*AR* Androgen Receptor, *ER* Estrogen Receptor, *BTK* Bruton's Tyrosine Kinase, *NTRK* Neurotrophic Tyrosine Receptor Kinase.

Fig. 2 | Schematic showing examples of spatio-temporal influence on PROTAC activity. **A** Spatial regulation of PROTAC-mediated AURKA degradation in mitotic cells. **B** Temporal regulation of PROTAC-mediated CDK2 degradation through the cell cycle. Created in BioRender. Cardno, A. (2025) https://BioRender.com/ud1xchr.

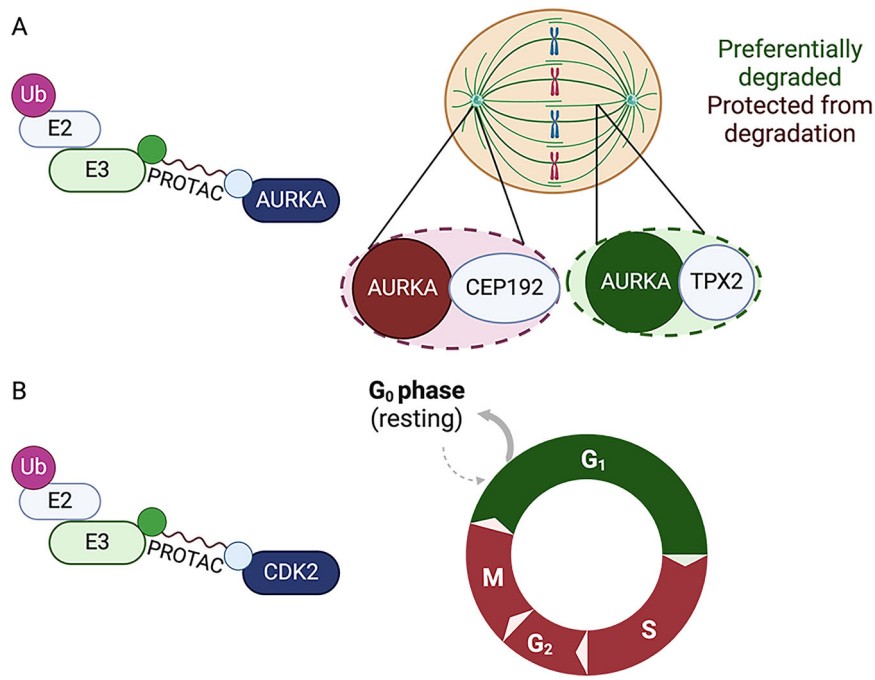

## Target-relevant parameters for PROTAC efficacy

number of PROTAC studies are starting to identify such factors, which either prevent or enhance target degradation, and in some cases indicate that TCF is not always the rate-limiting step. Here we review some of the cellular parameters demonstrated to govern outcomes of PROTAC interventions, or likely to be implicated in future as we learn more of the cell biology of TPD. Such information will be critical for the design of novel PROTAC strategies. While the scope of this review is confined to PROTAC discovery and research, many of the parameters influencing PROTAC efficacy may also apply to molecular glues and other heterobifunctional modalities.

### Target-relevant parameters for PROTAC efficacy

Many targets of clinical interest perform distinct functions depending on their subcellular localisation. Some studies have shown that localisation of a target is an important determinant of its susceptibility to PROTAC-mediated degradation.

In an effort to understand the 'PROTACtable genome', Schneider et al. developed a PROTACtability workflow to predict the viability of PROTAC targeting for a given protein of interest[27]. Cellular location of the target based upon UniProt and GO location data[28] was one factor considered. Targets categorised as cytoplasmic/cytosol- or nuclear-localised were considered 'good' for PROTAC development while targets localising in other compartments were assigned a lower confidence score. These categorisations were based upon the expectation that essential ubiquitination machinery is cytoplasmic or nuclear. Nevertheless, PROTACs against targets that do *not* localise to the cytoplasm or the nucleus have been successfully developed, for example for integral membrane proteins such as EGFR and PD-L1[29]. These examples illustrate that we still have much to learn about the spatial organisation of ubiquitin signalling pathways.

An experimental study by Simpson et al. explored the influence of target localisation on PROTAC-induced degradation using a tag-based approach; dTAG PROTACs were used to recruit VHL and CRBN to FKBP12[F36V], and the VHL-recruiting HaloPROTAC was employed to degrade the same substrate via a Halo tag. Various peptide localisation signals were attached to FKBP12[F36V]/Halo to evaluate how subcellular localisation of the target affected its susceptibility to the different PROTACs[30]. Degradation efficiency of dTAG[VHL] was superior to dTAG[CRBN] in the endoplasmic reticulum whilst dTAG[CRBN] performed better than dTAG[VHL] towards nuclear, cytoplasmic, and mitochondrial targets. In

general, the dTAG PROTACs appeared to be most active in nuclear and cytoplasmic compartments and least active in Golgi, peroxisome, and lysosomal compartments. The HaloPROTAC showed a broad spectrum of activity across different subcellular compartments, excluding the Golgi compartment. In this study the differential sensitivity of a target to the same PROTAC was not directly compared across cellular compartments. However, the study does highlight that the same target is degraded to differing extents by PROTACs recruiting different E3s and that cellular compartments are a source of this variability.

Characterisation of AURKA-targeting PROTACs revealed a functionally relevant example of localisation-dependent PROTAC-mediated degradation[31]. AURKA plays an important role in mitosis where it localises to the centrosomes and mitotic spindle, facilitating centrosome maturation and spindle assembly[32]. PROTAC-D, a CRBN-recruiting PROTAC that targets AURKA via the MLN8237 (alisertib) ligand, was shown to selectively degrade AURKA on the mitotic spindle, whilst leaving the centrosomal pool of active AURKA intact (Fig. 2A). Knockdown of CEP192, the binding partner required for AURKA localisation at the centrosome[33], rendered AURKA more sensitive to degradation by PROTAC-D[31].

In this example, whether differential PROTAC sensitivity is due to reduced binding of PROTAC to centrosomal AURKA, steric hindrance of TCF or post-ubiquitination 'rescue' of target protein, remains to be elucidated. However, the centrosome is a highly dense structure - described in some studies as a biomolecular condensate arising from phase separation[34,35] - that acts to selectively segregate cellular components and may exclude PROTACs or prevent recruitment of E3 complexes. Irrespective of the composition of the centrosome, it would also seem inevitable that binding partners of target proteins – such as the AURKA binding partners CEP192 and TPX2 – should play a role in controlling the access of ubiquitination machinery recruited by PROTACs, perhaps through conformational effects on target engagement with the PROTAC or E3, or through steric hindrance of E3 recruitment, or through physical occlusion of lysine residues required for ubiquitination.

Regulation of PROTAC sensitivity by target binding partners could also give rise to temporal differences in sensitivity, where binding partners are regulated, or change, over time. For example, CDK2, another cell cycle-regulated kinase, has been shown to be differentially sensitive to PROTAC-mediated degradation depending on cell cycle phase. CDK2 was most

efficiently degraded by CRBN-recruiting pan-kinase degrader TL12-186 in serum-starved (G0) cells and not degraded at all in S-phase or M-phase arrested cells, despite the PROTAC binding to CDK2 under all conditions[36] (Fig. 2B). TCF was found to occur only in serum-starved cells, and the authors hypothesised that the presence of CDK2 in complex with other interactors (Cip/Kip cyclin-dependent kinase inhibitors) during other phases of the cell cycle[37] could sterically prevent CRBN recruitment and ubiquitination of CDK2.

In a recent screen of chemical inhibitors influencing PROTAC-mediated degradation of BRD4 transcription factors, Mori et al. identified the poly-ADP ribosylation pathway as a substrate-specific ubiquitination enhancer, proposing that inhibition of this signalling pathway promotes the dissociation of BRD factors from chromatin, consistent with the idea that the presence of such factors in biologically active complexes is limiting for PROTAC engagement or TCF[38].

Given the differential sensitivity of PROTAC targets based upon their presence in biological complexes, it is inevitable that target expression levels can influence PROTAC efficacy. Although some studies have found that exogenously expressed target protein is poorly degraded[39], others have observed that overexpressed targets are degraded better than endogenously expressed targets, likely due to a higher abundance of 'free' target protein. For example, when comparing the degradation of exogenously expressed HiBiT-tagged WDR5 and endogenous WDR5, DCAF1-recruiting PRO-TACs showed up to two-fold higher $D_{max}$ values for degradation of the exogenous WDR5[40]. Another recent study demonstrated a similar phenomenon, whereby endogenous targets ITK1 and AURKA were weakly degraded by VHL-recruiting PROTACs, as measured by western blot, whereas ectopically expressed HiBiT-tagged substrates were degraded to a greater extent than the endogenous substrate in a dose-dependent manner[41]. The authors suggested that altered subcellular localisation of ectopically expressed substrates may lead to differences in PROTAC accessibility and therefore degradability. These observations may provide an advantageous application of PROTACs in diseases where targets are overexpressed.

Overall, these studies indicate that the subcellular context of targets, including localisation and presence in multi-protein complexes, can have a profound impact on their susceptibility to PROTAC-mediated degradation with sensitivity of TCF to cellular factors a key consideration.

Experimental and computational approaches are beginning to inform our understanding of TCF. For example, the crystal structure of BRD4 degrader MZ1 in ternary complex with the BRD4 bromodomain (BRD4$^{BD2}$) and VHL (PDB 5T35) has provided insight into mechanisms of selectivity and cooperativity of MZ1[7]. Recent crystal structures of DCAF1-PROTAC-WDR5 ternary complexes (PDB 9B9H, 9B9T, 9B9W, 9BA2, 9DLW) identified essential DCAF1 loops required for surface plasticity, and yielded insights into the need for an optimal linker length to promote a stable and productive ternary complex[40]. The structure of SMARCA2 in complex with VHL and a bivalent PROTAC compound (PDB 6HAY) has allowed a structure-guided approach to optimising a potent and cooperative SMARCA2 degrader, ACBI1[42]. Another study described crystal structures of degrader-mediated BTK and cIAP1 ternary complexes (PDB 6W8I, 8DSO) that were used for structure-based design of a PROTAC linker more favourable to TCF. Interestingly, this study concluded that increased ternary complex stability did not correlate with increased degradation efficiency[43], presumably due to suboptimal orientation of target and E3 or loss of flexibility required for later steps in the ubiquitination pathway.

Structural studies also enable identification of putative PROTAC-induced sites of ubiquitination on target proteins. In one such study, Bai et al. used Rosetta software[44] in developing a computational workflow to predict TCF, modelled using CRBN-recruiting PROTACs against a number of CDKs[45]. The model was then validated by site-directed mutagenesis of lysine residues predicted to act as ubiquitin receptors. Importantly, some ternary complex orientations were proposed to be 'unproductive' due to suboptimal positioning of lysines, despite stable TCF[45], highlighting availability of ubiquitin receptors as a key parameter in PROTAC sensitivity.

Many challenges remain in trying to produce structures of ternary complexes able to predict ubiquitination. The large size of E3 multi-protein complexes means that many experimental models to date have been restricted to the substrate receptor/adaptor components. The flexible nature of ternary complexes induced by small molecule PROTACs cannot always be captured by computationally predicted models. Furthermore, 3D structures will often be based upon the free target protein and do not represent the biologically active version, which is likely to exist in complex with binding partners that vary according to localisation, cell cycle stage, disease state and is subject to post-translational modifications. All these substrate-specific parameters are likely to influence target conformation, ternary complex assembly, accessibility of lysine residues and ubiquitin transfer.

## UPS machinery influencing PROTAC efficacy

The expression level and localisation of E3 ubiquitin ligases and their cognate E2s is undoubtedly a key parameter in PROTAC effectiveness, although one for which limited information is available. Considering the example of CRBN: it has been shown by immunofluorescence to localise to both the cytoplasm and the nucleus[46,47]. The nuclear localisation of CRBN is required for regulating Ikaros transcriptional activity in response to thalidomide[48] and more recently, a genome-wide screen for factors affecting the anti-myeloma activity of pomalidomide identified Karopherin beta 1 (KPNB1), a nuclear import protein required for nuclear import of CRBN and CRBN-dependent degradation of IKZF3[46]. Conversely cytoplasmic localisation of CRBN was required for CC-885 (a thalidomide derivative) to induce degradation of cytoplasmic translation factor GSPT1[46]. These studies highlight the overlap of target and E3 subcellular spatial distribution as a factor in effective proximity-induced degradation.

In illustration of how such spatial information can be exploited, a screen for PROTACs covalently binding E3s via cysteine-reactive electrophilic ligands identified DCAF16, a poorly characterised substrate receptor component of CUL4-DDB1. Engagement of DCAF16 by the electrophilic fragment promoted selective degradation of an NLS-tagged substrate, NLS-FKBP12[49]. The demonstrated nuclear localisation of DCAF16 makes it an attractive E3 to use for PROTAC strategies aiming to deplete nuclear-localised proteins. To evaluate the effectiveness of such a strategy, the DCAF16-binding electrophilic fragment was synthesised into a PROTAC against BRD4, a nuclear transcription factor and regulator of many oncogenes including Myc[50], using the potent and selective BRD4-binding compound JQ1. The PROTAC KB02-JQ1 successfully degraded BRD4 in a DCAF16- and proteasome-dependent manner.

Variation in expression levels of E3s will influence their utility in PROTAC strategies. Liu et al. investigated degradation of one specific target, AURKA, in response to a panel of PROTACs recruiting three different E3s – CRBN, VHL and cIAP1[51]. Their study showed that the sensitivity of AURKA degradation to different E3s was dependent on cell cycle phase. The CRBN-recruiting PROTAC preferentially degraded mitotic pools of AURKA, while the cIAP1 PROTAC was better at degrading interphase AURKA in acute myeloid leukaemia cells. The authors showed cell-cycle dependent changes in the expression levels of CRBN and cIAP1 and proposed this as a mechanism to explain temporal differences in target degradation capacity.

Recently, a review of the variable response of multiple myeloma patients to IMiDs highlighted a correlation with CRBN expression levels[52]. One study in particular showed that patients with low levels of CRBN showed limited clinical response to IMiDs whilst those with higher levels of CRBN showed a partial or improved response[53]. It is too soon to know how clinical outcomes of PROTAC treatments correlate with E3 expression, but it seems likely that in future, similar studies exploring the correlation of E3 expression and patient response to PROTACs will facilitate the use of expression of CRBN and other E3s as biomarkers for stratification of patient populations for therapy. Indeed, future strategies can seek to exploit differences in E3 expression to design tissue-specific PROTAC strategies. For

example, a recently described novel PROTAC, ABT-263, was able to recruit VHL for destruction of anti-apoptotic factors BCL-X$_L$/BCL-2 whilst sparing the platelet toxicity usually seen as on-target side effect of BCL-X$_L$/BCL-2 inhibition, as platelets do not express VHL[54].

Variation of E3 expression is also likely to be a key factor in acquired resistance to PROTACs: A genome-wide CRISPR-Cas9 screen in myeloma cells treated with PROTACs for various targets showed that it was loss of function of CRBN/VHL, rather than 'work-around' mechanisms for the loss of the respective oncoprotein target, which underpinned resistance to degradation[55]. This finding corroborated earlier results from Zhang et al. who showed that chronic treatment of cancer cells with BET-PROTACs led to genomic alterations in the core components of the E3 complexes, rather than acquired mutations in the target protein[56]. These studies highlight the importance of E3 expression patterns in determining PROTAC efficacy and provide strong rationale to expanding the E3 landscape to provide options for overcoming resistance.

The steps which follow recruitment of neosubstrate and E3 into a ternary complex, and which determine the subsequent proteolysis event, have been relatively understudied, as recently illustrated by the finding that HSP90 inhibition promotes degradation of PROTAC-targeted BRD4 independently of ubiquitination[38]. Ubiquitin chain architecture is critical in determining the fate of a ubiquitinated protein according to the ubiquitin code, yet there is still much to be discovered both about the nature of the code and about how ubiquitin chains are assembled on PROTAC-targeted neosubstrates. This gap in knowledge has been highlighted by recent discoveries of roles for additional ubiquitinating machinery – beyond the PROTAC-recruited E3 – in PROTAC-mediated degradation (Fig. 3). A model for the formation of branched K29/K48 ubiquitin chains on BRD4 induced by VHL-recruiting PROTAC MZ1 was elegantly demonstrated using in vitro ubiquitination assays in combination with mass spectrometry[57]. Not only did the authors show the existence of branched K29/K48 ubiquitin chains, but they also uncovered a role for TRIP12, a HECT-type E3, in facilitating the addition of K29 linkages onto pre-existing K48-linked chains. The cooperativity of two E3s in assembling ubiquitin chains facilitated rapid degradation of BRD4 by MZ1. To explain the lack of TRIP12-dependence for degradation of the endogenous VHL substrate HIF1α, the authors argued that E3 cooperativity may be important for driving degradation of neosubstrates by providing additional ubiquitin conjugation capacity to reactions where the pairing of substrate and E3 is not evolutionarily optimised.

In a more recent study, branched ubiquitin architecture was identified as an output of cIAP1-recruiting degraders, unveiling a novel ubiquitin chain-type that facilitates proteasomal degradation in the context of targeted protein degraders. The K63-specific E2 enzyme UBE2N was shown to be required for degradation of cIAP1 neosubstrates through conjugation of branched K11/48 linkages onto a K63-linked chain, supporting a model whereby branched linkages are exposed at the distal surfaces of ubiquitin chains and efficiently increase the density of ubiquitin signals[58].

A number of studies have highlighted the role of branched chains in efficient proteolysis of ubiquitin-conjugates, thought to be as a result of high ubiquitin density for binding ubiquitin receptors on the 19S regulatory particle of the proteasome[2,59,60]. The involvement of additional E2 and E3 enzymes, including cooperative E3s, sometimes termed 'E4' activities, could aid proteolysis in specific cellular compartments. Nuclear-specific E3 enzymes, such as TRIP12[57] and compartmented activity of E4 enzymes like UBE4B and p300/CBP[61–63] illustrate how PROTACs could be optimised to target distinct protein pools in subcellular localisations, offering promising strategies to deplete specific pools of target proteins which show disease-relevant functions.

In order to exploit the enhanced degradation efficiency associated with different ubiquitin chain architectures, PROTACs could be rationally designed to recruit E3s known to synthesise branched chains. For example, CRL complexes SCF$^{β-TRCP}$ and SCF$^{FBXW7}$ can catalyse branched K11/K48-linked chains, while HECT-type E3 ligases HUWE1 and UBR5 can generate mixed K48/K63-linked chains; both branched chain types leading to enhanced proteasomal degradation efficiency[64]. Alternatively, the emergence of multi-valent PROTACs such as SIM1, which recruits VHL to BRD2 via two bromodomain binding sites[65], and AB3076, which recruits CRBN and VHL to BET proteins to induce additive ubiquitin chain formation[66], provide proof of concept that PROTACs can be rationally designed to recruit more than two components. Such strategies might enable design of more efficient PROTACs through recruitment of E3 and/or E2 combinations favouring branched ubiquitin topologies.

The machinery required for building ubiquitin chains exists alongside other components of the UPS that modulate ubiquitin chain size and topology. Deubiquitinases (DUBs) play a critical role in removing or editing ubiquitin chains[67]. There are over 100 DUBs characterised into seven families. The USP family often lack chain specificity[68] whilst other DUB families exhibit highly selective activity[69]. It seems highly likely that DUBs

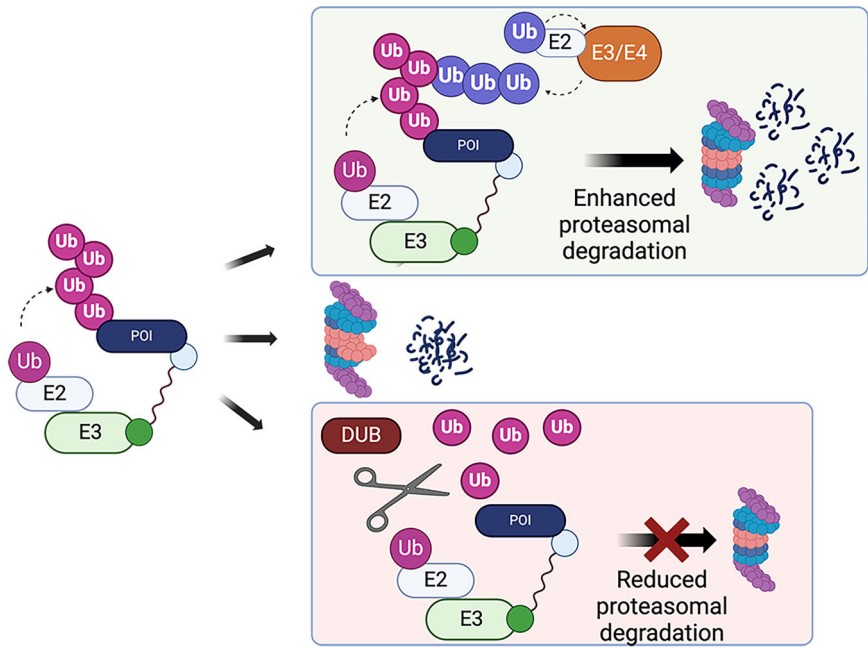

**Fig. 3 | Schematic of cellular factors influencing PROTAC activity.** Additional ubiquitin ligase activities (E3/E4) can act to promote ubiquitination and degradation of substrates in certain contexts (top) while deubiquitinases (DUBs) can remove or trim ubiquitin conjugates to stabilise or 'rescue' neosubstrates from degradation at the proteasome (bottom). Created in BioRender. Cardno, A. (2025) https://BioRender.com/zozyz3i.

act to 'rescue' targets from degradation by limiting the effectiveness of PROTAC-mediated ubiquitination against a given target or in a particular subcellular compartment. Indeed, systematic analysis of the subcellular localisation of over 60 DUBs highlighted the variety in expression patterns of these enzymes[70]. More recently, USP15 has been shown to antagonise lenalidomide-mediated degradation of CRBN neo-substrates including IKZF3 and CK1α and PROTAC dBET1-targeted BRD4[71]. USP15 was also shown to be highly expressed in IMiD-resistant cell lines[71], pointing to DUB status as a key factor to consider in stratification of cancer patient populations for future PROTAC-based therapy.

Finally, the onwards processing of a ubiquitinated protein involves numerous parameters for which investigation so far has been limited. For example, proteasomal degradation of many substrates relies on the ATP-dependent extraction of ubiquitinated target from cellular complexes by the Cdc48/p97 'segregase'[72]. A study by van Nguyen et al. investigated the role of p97 in the degradation of substrates targeted by CRL4$^{CRBN}$, including glutamine synthetase and IMiD neosubstrates IKZF1, IKZF3, CK1α and GSPT1[73]. p97 promoted the disassembly of ubiquitinated glutamine synthetase subunits from its homodecameric complex, enabling efficient proteasomal degradation. Furthermore, p97 was essential for degradation of CRBN-dependent IMiD neosubstrates. These findings highlight the potential of p97-dependent substrate processing to be a key factor influencing PROTAC efficacy, at least for CRBN-recruiting PROTACs.

Following processing by p97, substrates are delivered to the proteasome by 'shuttling factors' that bind simultaneously to ubiquitinated substrates and to the proteasome[72]. Substrates docked at the ubiquitin receptor subunits of the 19S regulatory particle can then be unfolded, provided they possess an intrinsically disordered region accessible to the ATPases guarding the entrance to the 20S core particle[74]. Simultaneous removal of ubiquitin conjugates enables the passage of the unfolded substrate into the catalytic core for digestion by the proteasomal peptidases[72]. Despite the abundance of proteasomes in the cell, unresolved questions around the localisation, transport and activity of proteasomes may be relevant to design of TPD strategies. Proteasomes localise in both the nucleus and cytoplasm, but the distribution is variable between cell types and cellular conditions. Studies of spatiotemporal dynamics of proteasomes using endogenously tagged subunits have led to an emerging consensus that there is an enrichment of 26S proteasome in the nucleus, with dynamic movement between nucleus and cytoplasm[75,76]. The recent identification of AKIRIN2 as a critical factor in proteasome nuclear import[77] underscores the importance of spatiotemporal regulation of proteasome activity and reveals gaps in our understanding of proteasome dynamics in cellular degradation pathways.

## Perspective

The undoubted therapeutic potential of TPD strategies has given rise to intense efforts to uncover new tools harnessing more of the cellular ubiquitination machinery, despite remarkable gaps in our knowledge of exactly which machinery contributes to PROTAC activity in vivo. Thus, many aspects of the cellular pathways required for successful target proteolysis remain incompletely described, even for those PROTACs shown to be highly effective in both pre-clinical and clinical settings.

Prioritising cellular assays measuring TCF and ubiquitination steps will be necessary for better understanding of the parameters discussed in this review and their impact on PROTAC efficacy. Techniques applicable to measuring protein-protein interactions in cellulo, such as NanoBRET[39] or FRET[78] are emerging as valuable tools to characterise TCF and ubiquitination kinetics, whilst recent developments in mass spectrometry are providing new insights into ubiquitin linkage topologies[57,58,79]. NanoBRET assays in particular enable high-throughput compound screening and further optimisation of such assays would enhance the mechanistic understanding of each step in the TPD cascade.

In summary, PROTACs are providing powerful investigational tools with which to probe the cellular pathways that determine the fate of ubiquitinated proteins; better knowledge of such pathways will guide the rational design of new PROTACs, informing the choice of components for therapeutic strategies that promise disease-specific target degradation. At the same time, by providing a means for rapid chemical knockdown of desired targets, PROTACs promise to become important tools in cell biology.

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

## Acknowledgements

We thank past and present members of the lab for enriching discussions about the cell biology of PROTACs. Work in C.L.'s lab is supported by Biotechnology and Biological Sciences Research Council (BBSRC) [grant no. BB/X007499/1]. A.C. is supported by a Studentship from AstraZeneca.

## Author contributions

C.L. and A.C. conceived and planned the review. A.C. reviewed the literature and wrote the manuscript. B.K contributed some ideas to an early draft. A.C and C.L. revised and edited the final version of the manuscript.

## Competing interests

The authors declare no competing interests.
