## [Peer review file · Communications Biology]

Cellular Parameters Shaping Pathways of Targeted Protein Degradation

Corresponding Author: Dr Catherine Lindon

Version 0:

Reviewer comments:

Reviewer #1

(Remarks to the Author)

Brief summary of the manuscript:

This perspective article shines a spotlight on several underappreciated cellular factors that can have a significant impact on PROTAC efficacy. In particular, the authors focus on three primary areas: target-relevant parameters, E3-relevant parameters, and ubiquitin conjugate processing. Supported by references from the literature, the authors make the case that target localization within cellular compartments or as part of multi-protein complexes can profoundly affect PROTAC efficacy. Because ternary complex formation and successful target protein ubiquitination rely upon adoption of specific conformations and the availability of lysine residues, it is critical to understand how assembly of intracellular complexes and post-translational modifications can influence these factors. Furthermore, the expression level and location of E3 ligases and associated UPS components will likewise affect ternary complex formation and subsequent proteasomal degradation of the protein following PROTAC treatment. Finally, the authors raise the interesting prospect for how alterations in ubiquitin chain architecture and processing can, depending on the cellular context, either enhance efficacy or lead to treatment resistance.

Overall impression of the work:

The article provides a well-written summary of how an improved understanding of the cellular context for targeted protein degradation will improve PROTAC design and the success of this modality in specific patient populations. This aspect is not widely considered in targeted protein degradation, and the field would broadly benefit from increased awareness of these cellular factors. Because there are many nuances related to cellular context that can impact PROTAC efficacy, the relatively concise focus of this perspective helps readers more easily digest the implications of the selected factors. The utility of these recommendations may be further enhanced if the authors describe how specific assays or experimental workflows investigating these cellular factors can be translated into actionable insights for PROTAC design.

Specific comments/recommendations:

1. The article does an excellent job making the case for why it is important to consider the cellular context for PROTAC efficacy. However, there is relatively limited discussion of how these factors can be interrogated experimentally in the context of typical PROTAC discovery and development. While tag-based approaches, imaging studies, or immunoprecipitation experiments can help to tease apart subcellular localization and complex formation in a given cellular context, such approaches may not be feasible or practical in many scenarios. The authors should clearly provide practical recommendations for what additional assays should be considered across PROTAC programs to study these cellular factors and what observations should be used to justify these more detailed (and resource-intensive) investigations.
2. The authors raise the interesting possibility that the choice of different target-binding warheads can influence target localization and thereby PROTAC efficacy. The chosen example related to AR-targeted PROTACs and the observation that different anti-androgens may alter translocation of cytoplasmic AR to the nucleus. Presumably this effect may extend beyond AR where other target-specific ligands may differentially influence target trafficking and subcellular localization. Additional examples of this effect (even for target-ligand pairs that may be relevant to PROTACs but not yet evaluated as PROTACs) would be valuable to demonstrate the potential breadth of this effect across targets.
3. The authors mentioned that variation in the expression levels of E3s can affect sensitivity to PROTAC treatment and even identify patients less likely to respond to a given PROTAC treatment. The authors should discuss whether it is likely to be

possible to prospectively establish expression level thresholds or whether this can only be accomplished retrospectively. Also, the authors do not mention how differences in target expression can influence PROTAC efficacy. Prior publications have shown differences in apparent PROTAC efficacy for degrading endogenously expressed targets vs. overexpressed targets. A few comments about the impact of target expression levels along with associated references would be helpful to include in the section on target-relevant factors.

4. Ubiquitin chain architecture was mentioned as an understudied aspect of PROTAC design, and examples with the VHL-based PROTAC, MZ1, and a cIAP1-based PROTAC were provided to show how different branched ubiquitin linkages can be produced. However, there was limited discussion of how alteration of the ubiquitin chain architecture can be accomplished through PROTAC design. If possible, it would be helpful to include a short discussion for how PROTACs can be rationally designed to leverage specific ubiquitin chain architectures, whether that is through recruitment of unique E3 ligases, adoption of specific ternary complex structures, or other mechanisms.

5. The authors underscore the potential contribution of DUB activity to counteract the activity of PROTACs, suggesting that DUB status could be a biomarker for patient-stratification. The one example provided relates to USP15. It would be helpful to include additional examples, if available, demonstrating how DUB activity can decrease PROTAC efficacy or how inhibition of DUB activity can enhance PROTAC efficacy. This would highlight the practical use of quantitative measures for DUB activity.

Reviewer #2

(Remarks to the Author)

Proteolysis-targeting Chimeras (PROTACs) are bifunctional small molecules which simultaneously recruit a target protein and an E3 ligase. PROTAC-induced proximity of E3 and target results in ubiquitination and proteasomal degradation of the target. By this degradative mechanism of action, PROTACs have attained significant interest in the drug discovery and chemical biology communities, both as potential therapies and as chemical tools for protein knockdown.

This manuscript is a review wherein the authors discuss the cellular influences on PROTAC performance. With reference to selected published works, the authors discuss the influence of several factors - target/E3 localisation and expression patterns, E3 activity, deubiquitinases (DUBs), and additional components of the ubiquitin-proteasome system (UPS) - on the sensitivity of a target to PROTAC-mediated degradation.

The manuscript is well-written, and the referenced works are relevant and of significant interest for those interested in PROTAC activity and development. The review covers a sufficiently broad scope of topics, but is written succinctly, enabling accessibility for readers with both limited and deep experience of the PROTAC field. I thoroughly enjoyed reading this review and thank the authors for putting it together!

Kind regards,
Nicole Trainor

Specific comments and recommendations:

1. Page 2 – ARV-110 has now been substituted for a second-generation molecule (ARV-766) in Phase III (<https://ir.arvinas.com/news-releases/news-release-details/potential-arvinas-protac-ar-degraders-reinforced-111-months>) The authors may wish to simply substitute mention of ARV-110 for ARV-766 in the manuscript or add ARV-766 to the sentence. A journal reference for ARV-766 performance in Phase II trials can be found here: https://doi.org/10.1200/JCO.2023.41.6_suppl.TPS290 ARV-766 was also highlighted in this opinion piece doi: 10.1186/s13046-024-03125-5.

2. Page 3 – Mention of a “stable” ternary complex. Substitution of “stable” with “productive” is recommended. Use of the word “stable” may imply a degree of favourability or cooperativity, which does not account for instances where less stable complexes (presumably fleeting or present in minor concentrations) have proven to be productive. For example: these BTK degraders 10.1038/s41589-020-00686-2, or in this study of CRBN-based BRD4 degraders 10.1038/s41589-018-0055-y. The authors may wish to expand on this point with these references.

3. Page 3 – mention of ARV-110 (see comment 1)

4. Page 6 – capital letter typo after “Mori et al.”

5. Page 6 – “the structure of SMARCA2 in complex with VHL and a bivalent predicted PROTAC”. Use of the word “predicted” is confusing here, as this PROTAC is definitively bivalent by its design and by virtue of it forming the ternary complex in the crystallography experiment. Removal of the word “predicted” is recommended.

6. Page 6 (as with comment 5) - To clarify the difference between experimental and theoretical ternary complex examples cited here, the authors could add the PDB accession codes for the ternary complexes solved by X-ray crystallography which are mentioned in this paragraph. VCB-MZ1-BRD4 (PDB 5T35), VCB-PROTAC 1-SMARCA2 (PDB 6HAY). At the end of this paragraph, it is recommended to name the optimised SMARCA2 degrader from the cited Farnaby et al. paper: e.g. “to optimise to the potent and cooperative SMARCA2 degrader, ACBI1[ref 37]”

7. Page 7 – “...to predict ubiquitination: The large...” – recommend substitution of this colon with a full stop.

8. Page 9 – “In a more recent study[recommend comma here]”

9. Page 11 – Discussion of onwards processing of a ubiquitinated protein, particularly in the context of neosubstrates – the authors may find this paper relevant to cite here <https://doi.org/10.1073/pnas.1700949114>. This study investigates the role of p97 in IMiD-induced degradation of CRBN neosubstrates (see fig 6). The authors are encouraged to review this work and, if appropriate, discuss it briefly in this paragraph.

Reviewer #3

(Remarks to the Author)

Dear Editor,

My apologies for the delay in the revision.
As follows my comments on the manuscript submitted

Best regards

Version 1:

Reviewer comments:

Reviewer #1

(Remarks to the Author)

The authors have done an excellent job incorporating the feedback from the initial reviews. These revisions have significantly strengthened the manuscript in multiple areas.

Two final considerations for the authors are provided below:

1. In the new paragraph discussing the potential impact of endogenous vs. ectopic target expression, the authors suggest that overexpressed targets are often degraded better than endogenously expressed targets. While the examples provided do support this point, it may be helpful to also point out examples where overexpression can impair degradability if the expression levels reach a high enough level to overwhelm the capacity of potential PROTAC-induced degradation. For example, Riching et al. described much lower degradation of exogenously overexpressed BRD4 than the endogenous HiBiT-BRD4 generated by CRISPR:

Riching KM, Mahan S, Corona CR, McDougall M, Vasta JD, Robers MB, Urh M, Daniels DL. Quantitative Live-Cell Kinetic Degradation and Mechanistic Profiling of PROTAC Mode of Action. *ACS Chem Biol*. 2018 Sep 21;13(9):2758-2770. doi: 10.1021/acscchembio.8b00692. Epub 2018 Aug 30. PMID: 30137962.

2. The authors added a comment that the higher target expression levels would also reduce the hook effect. The hook effect is likely to be influenced by the E3 expression levels, but this is not necessarily significantly influenced by target expression levels (for example, even at low target expression levels, saturation of the target with excess available PROTAC should still drive maximal degradation as long as E3 is not limiting). However, target expression levels (and subcellular localization) can influence the observed Dmax, and this point is already emphasized in the article. To avoid any potential confusion about the impact of target levels on the hook effect, the sentence could simply be removed.

Reviewer #2

(Remarks to the Author)

Many thanks for your consideration of the suggested edits. I recommend no further changes at this stage.

Best regards,
Nicole Trainor

Reviewer #3

(Remarks to the Author)

The paper submitted by Lindon and collaborators has been substantially improved after deep revision. The reviewers' suggestions and comments were mostly addressed, resulting in a more easily readable review that is now better structured and organised. The material was better organised and the whole structure and content are more consistent across the paper.

We thank the reviewers for their insightful commentary and specific recommendations for improvements to our manuscript. We have revised the manuscript accordingly, as detailed in our point-by-point responses below.

Reviewer #1

1. The article does an excellent job making the case for why it is important to consider the cellular context for PROTAC efficacy. However, there is relatively limited discussion of how these factors can be interrogated experimentally in the context of typical PROTAC discovery and development. While tag-based approaches, imaging studies, or immunoprecipitation experiments can help to tease apart subcellular localization and complex formation in a given cellular context, such approaches may not be feasible or practical in many scenarios. The authors should clearly provide practical recommendations for what additional assays should be considered across PROTAC programs to study these cellular factors and what observations should be used to justify these more detailed (and resource-intensive) investigations.

We appreciate the reviewer's suggestion to provide more practical recommendations regarding additional assays for studying cellular factors in PROTAC programmes. We have added a paragraph to the conclusion outlining specific assays that can be employed in the early stages of PROTAC programmes to investigate key steps in the TPD cascade such as TCF and ubiquitination. We believe these additions will offer clearer, actionable insights for researchers designing PROTAC studies. The paragraph (on p13) reads:

“Prioritising cellular assays measuring TCF and ubiquitination steps will be necessary for better understanding of the parameters discussed in this review and their impact on PROTAC efficacy. Techniques applicable to measuring protein-protein interactions in cellulo, such as NanoBRET⁷⁸ or FRET⁷⁹ are emerging as valuable tools to characterise TCF and ubiquitination kinetics, whilst recent developments in mass spectrometry are providing new insights into ubiquitin linkage topologies^{56,57,80}. NanoBRET assays in particular enable high-throughput compound screening and further optimisation of such assays would enhance the mechanistic understanding of each step in the TPD cascade.”

2. The authors raise the interesting possibility that the choice of different target-binding warheads can influence target localization and thereby PROTAC efficacy. The chosen example related to AR-targeted PROTACs and the observation that different anti-androgens may alter translocation of cytoplasmic AR to the nucleus. Presumably this effect may extend beyond AR where other target-specific ligands may differentially influence target trafficking and subcellular localization. Additional examples of this effect (even for target-ligand pairs that may be relevant to PROTACs but not yet evaluated as PROTACs) would be valuable to demonstrate the potential breadth of this effect across targets.

We have added in a further example (the EGFR inhibitor AG-1478 influences cellular localisation of EGFR) to strengthen the argument that the choice of target ligand may influence the success of the PROTAC for targets which are susceptible to trafficking in response to ligands. The text (p3) now reads:

“Other ligands (and potential PROTAC warheads) similarly act to modulate cellular behaviour of their targets, for example, the EGFR inhibitor AG-1478 prevents the nuclear translocation of EGFR stimulated by EGFR ligand shB-EGF, trapping EGFR in the cytoplasm and at the plasma membrane²⁶. The choice of target binder in PROTAC design may therefore influence localisation of TCF, potentially a critical parameter in determining the degradability of neosubstrates.”

3. The authors mentioned that variation in the expression levels of E3s can affect sensitivity to PROTAC treatment and even identify patients less likely to respond to a given PROTAC treatment. The authors should discuss whether it is likely to be possible to prospectively establish expression level thresholds or whether this can only be accomplished retrospectively. Also, the authors do not mention how differences in target expression can influence PROTAC efficacy. Prior publications have shown differences in apparent PROTAC efficacy for degrading endogenously expressed targets vs. overexpressed targets. A few comments about the impact of target expression levels along with associated references would be helpful to include in the section on target-relevant factors.

With regards to the reviewer's first point, we have modified the sentence to more strongly make the point that future studies should investigate correlations between E3 expression and patient response to PROTACs, to enable such information to be used for patient stratification. On p9: *"... it seems likely that in future, similar studies exploring the correlation of E3 expression and patient response to PROTACs will facilitate the use of expression of CRBN and other E3s as biomarkers for stratification of patient populations for therapy"*

With regards to the second point about the influence of target expression levels on PROTAC efficacy, we found this a very useful comment and have now added a paragraph to highlight the observation that overexpressed targets are often better degraded than the endogenous target. This paragraph (p6/7) reads:

"Overexpressed targets are often degraded better than endogenously expressed targets, likely due to a higher abundance of 'free' target protein. Higher target expression levels would also reduce the 'hook effect' whereby excess PROTAC can inhibit TCF through binary engagement of target or E3. Indeed, when comparing the degradation of exogenously expressed HiBiT-tagged WDR5 and endogenous WDR5, DCAF1-recruiting PROTACs showed up to two-fold higher D_{max} values for degradation of the exogenous WDR5³⁹. Another recent study demonstrated a similar phenomenon, whereby endogenous targets ITK1 and AURKA were weakly degraded by VHL-recruiting PROTACs, as measured by western blot, whereas ectopically expressed HiBiT-tagged substrates were degraded to a greater extent than the endogenous substrate in a dose-dependent manner⁴⁰. The authors suggested that altered subcellular localisation of ectopically expressed substrates may lead to differences in PROTAC accessibility and therefore degradability. These observations may provide an advantageous application of PROTACs in diseases where targets are overexpressed."

4. Ubiquitin chain architecture was mentioned as an understudied aspect of PROTAC design, and examples with the VHL-based PROTAC, MZ1, and a cIAP1-based PROTAC were provided to show how different branched ubiquitin linkages can be produced. However, there was limited discussion of how alteration of the ubiquitin chain architecture can be accomplished through PROTAC design. If possible, it would be helpful to include a short discussion for how PROTACs can be rationally designed to leverage specific ubiquitin chain architectures, whether that is through recruitment of unique E3 ligases, adoption of specific ternary complex structures, or other mechanisms.

In response to this excellent point, we have added a paragraph detailing how PROTACs could be rationally designed to enhance degradation efficiency by inducing the formation of specific ubiquitin chain architectures. This paragraph (p11) reads:

"In order to exploit the enhanced degradation efficiency associated with different ubiquitin chain architectures, PROTACs could be rationally designed to recruit E3s known to synthesise branched chains. For example, CRL complexes $SCF^{\beta-TRCP}$ and SCF^{FBXW7} can catalyse branched K11/K48-linked

chains, while HECT-type E3 ligases *HUWE1* and *UBR5* can generate mixed K48/K63-linked chains; both branched chain types leading to enhanced proteasomal degradation efficiency⁶³. Alternatively, the emergence of multi-valent PROTACs such as *SIM1*, which recruits VHL to BRD2 via two bromodomain binding sites⁶⁴, and *AB3076*, which recruits CRBN and VHL to BET proteins to induce additive ubiquitin chain formation⁶⁵, provide proof of concept that PROTACs can be rationally designed to recruit more than two components. Such strategies might enable design of more efficient PROTACs through recruitment of E3 and/or E2 combinations favouring branched ubiquitin topologies.”

5. The authors underscore the potential contribution of DUB activity to counteract the activity of PROTACs, suggesting that DUB status could be a biomarker for patient-stratification. The one example provided relates to USP15. It would be helpful to include additional examples, if available, demonstrating how DUB activity can decrease PROTAC efficacy or how inhibition of DUB activity can enhance PROTAC efficacy. This would highlight the practical use of quantitative measures for DUB activity.

To our knowledge, there are no further examples in the literature (at present) of how DUB activity modulates PROTAC efficacy.

Reviewer #2

1. Page 2 – ARV-110 has now been substituted for a second-generation molecule (ARV-766) in Phase III (<https://ir.arvinas.com/news-releases/news-release-details/potential-arvinas-protacr-ar-degraders-reinforced-111-months>) The authors may wish to simply substitute mention of ARV-110 for ARV-766 in the manuscript or add ARV-766 to the sentence. A journal reference for ARV-766 performance in Phase II trials can be found here: https://doi.org/10.1200/JCO.2023.41.6_suppl.TPS290 ARV-766 was also highlighted in this opinion piece doi: 10.1186/s13046-024-03125-5.

We thank the reviewer for updating us on the progress of Arvinas PROTACs. We have modified the manuscript to mention ARV-766 alongside ARV-110.

2. Page 3 – Mention of a “stable” ternary complex. Substitution of “stable” with “productive” is recommended. Use of the word “stable” may imply a degree of favourability or cooperativity, which does not account for instances where less stable complexes (presumably fleeting or present in minor concentrations) have proven to be productive. For example: these BTK degraders 10.1038/s41589-020-00686-2, or in this study of CRBN-based BRD4 degraders 10.1038/s41589-018-0055-y. The authors may wish to expand on this point with these references.

Indeed, we acknowledge the accuracy of the reviewer’s observation and have replaced the word ‘stable’ with ‘productive’. The relevant sentence (p3) now reads:

“The composition and length of the linker between binding moieties for substrate and E3 strongly influence the activity of TPD tools by contributing to formation of a productive ternary complex which is cooperative in nature yet flexible enough to allow ubiquitination of targets“

We have added in discussion on the BTK degraders paper in the section on structural studies of TCF (p7) as follows:

“Another study described crystal structures of degrader-mediated BTK and cIAP1 ternary complexes (PDB 6W8I, 8DSO) that were used for structure-based design of a PROTAC linker more favourable to TCF. Interestingly, this study concluded that increased ternary complex stability did not correlate with increased degradation efficiency⁴², presumably due to suboptimal orientation of target and E3 or loss of flexibility required for later steps in the ubiquitination pathway. “

3. Page 3 – mention of ARV-110 (see comment 1)

We added mention of ARV-766 (which we assume is what was intended here!).

4. Page 6 – capital letter typo after “Mori et al.”

Typo now corrected.

5. Page 6 – “the structure of SMARCA2 in complex with VHL and a bivalent predicted PROTAC”. Use of the word “predicted” is confusing here, as this PROTAC is definitively bivalent by its design and by virtue of it forming the ternary complex in the crystallography experiment. Removal of the word “predicted” is recommended.

We agree – the word ‘predicted’ has been removed.

6. Page 6 (as with comment 5) – To clarify the difference between experimental and theoretical ternary complex examples cited here, the authors could add the PDB accession codes for the ternary complexes solved by X-ray crystallography which are mentioned in this paragraph. VCB-MZ1-BRD4 (PDB 5T35), VCB-PROTAC 1-SMARCA2 (PDB 6HAY). At the end of this paragraph, it is recommended to name the optimised SMARCA2 degrader from the cited Farnaby et al. paper: e.g. “to optimise to the potent and cooperative SMARCA2 degrader, ACBI1[ref 37]”

We thank the reviewer for providing the PDB codes and have added them to the manuscript where appropriate.

7. Page 7 – “...to predict ubiquitination: The large...” – recommend substitution of this colon with a full stop.

Colon has been substituted with a full stop to improve ease of reading.

8. Page 9 – “In a more recent study[recommend comma here]”

Comma introduced here.

9. Page 11 – Discussion of onwards processing of a ubiquitinated protein, particularly in the context of neosubstrates – the authors may find this paper relevant to cite

here <https://doi.org/10.1073/pnas.1700949114>. This study investigates the the role of p97 in IMiD-induced degradation of CRBN neosubstrates (see fig 6). The authors are encouraged to review this work and, if appropriate, discuss it briefly in this paragraph.

We agree this is an important paper to discuss and are not sure how we overlooked its inclusion. We have now added it to our manuscript (on p12), to strengthen the argument that ubiquitin-processing factors, such as p97, are of relevance and importance to success of TPD strategies:

“A study by van Nguyen et al. investigated the role of p97 in the degradation of substrates targeted by CRL4^{CRBN}, including glutamine synthetase and IMiD neosubstrates IKZF1, IKZF3, CK1 α and GSPT1⁷³. p97 promoted the disassembly of ubiquitinated glutamine synthetase subunits from its homodecameric complex, enabling efficient proteasomal degradation. Furthermore, p97 was essential for degradation of CRBN-dependent IMiD neosubstrates. These findings highlight the potential of p97-dependent substrate processing to be a key factor influencing PROTAC efficacy, at least for CRBN-recruiting PROTACs.”

Reviewer #3

- What a reader would expect from the title is "the effect of some cellular feature to the PROTACs activity, such as cellular hijack of the compound, cellular metabolism of the compound, efflux, or factors like these, which is not what is reported in your manuscript, so I would recommend you use a more detailed and precise title.

We're not sure why this particular interpretation would be placed on the title but have modified it to try and strengthen our intended meaning. We have therefore replaced the word 'influence' with 'parameter' and included the word 'pathways' to indicate that we are referring to the biological pathways of TPD action. The title now reads:

“Cellular Parameters Shaping Pathways of Targeted Protein Degradation”

- Page 3. The authors state "Ternary complex formation (TCF) is therefore widely assumed to be the rate-limiting step in TPD in vivo." suggesting this is just an assumption, while it is widely proved that TCF is ONE of the limiting steps of TPD in vivo. I suggest the sentence to be re-phrased.

We agree that TCF is *one of the possible limiting steps* for PROTAC action, under any conditions. However our understanding is that there can only be one actual rate limiting step, and there has been rather little investigation so far of what this step might be *in vivo*. We have rephrased the sentence (p3) in a way that at once acknowledges the gap in knowledge and accepts current thinking on this point. It now reads:

“This process has been shown in many studies to correlate with the efficiency of neosubstrate degradation^{6,21,22}. Ternary complex formation (TCF) is therefore widely accepted as the rate-limiting step in TPD in vivo. “

We hope this more acceptable to the reviewer.

- Page 3. The authors state "Screening of compounds for TPD activity when measures target protein degradation as the endpoint, bypassing steps between target engagement and degradation that may include the rate-limiting step of protein degradation for a particular target

in a particular cellular context. This means there is little information available to explain why compounds optimised in vitro may fail in in cellulo assays of target degradation". This is not completely true, for many PROTAC reported as products of drug discovery campaigns the ternary complex formation is measured and nowadays several techniques are available and used to do that. I would recommend that the authors resent this phrase.

We acknowledge that ternary complex is measured in some way for many PROTACs reported in drug discovery campaigns. However, the point we were making here is that early-stage screening of compounds often uses target degradation as an assay for hit selection. For compounds which do not show successful enough degradation to be taken forward in drug discovery pipelines, we have limited understanding of why they have failed – i.e., through TCF? Ubiquitination?

We have therefore amended the sentence by addition of the term 'early-stage screening' to make our point more clearly. This sentence (p3) now reads:

“Early-stage screening of compounds for TPD activity often measures target protein degradation as the endpoint, bypassing steps between target engagement and degradation that may include the rate-limiting step of protein degradation for a particular target in a particular cellular context.”

- Page 3. With "warhead" the authors mean the target binder? PROTACs have two "warheads", generally speaking, I would suggest they specify which of them they intentionally mean here.

We acknowledge the potential for ambiguity here and have replaced the term 'warhead' with 'target binder'.

- Page 7. In this reviewer's opinion, a review should report all the relevant available literature; here the challenges and difficulties in obtaining the TC 3D structures are stated, but the successful works in obtaining the 3D "Ternary" complexes are not even cited. I know the main scope of this review is to highlight the challenges we are still facing, but saying "Many challenges remain in trying to produce structures of ternary complexes able to predict ubiquitination: The large size of E3 multi-protein complexes means that most experimental models to date have been restricted to the substrate receptor/adaptor components" might sound to the reader that we basically have no structures of complexes, and it is not the case. I would suggest the authors to review the literature and re-shape this part of the paragraph accordingly.

We acknowledge the importance of citing all of the relevant literature. In addition to the relevant citations in the original manuscript (BRD4-MZ1-VHL ternary complex, SMARCA2-PROTAC-VHL ternary complex) we now also cite recent structures for the DCAF1-PROTAC-WDR5 ternary complexes and a study providing crystal structures of BTK-PROTAC-clAP1 ternary complexes (p7).

- Figures 1 & 3. The protein target is depicted as a disordered/unfolded protein, was it chosen on purpose?

If not, I would recommend the authors depict the protein in a structured form, more than an unstructured polypeptide, being a general representation of the PROTACs mechanism of action.

The protein target was not intentionally represented as an unfolded polypeptide, so figures have been amended to avoid this interpretation by representing the target proteins as globular.

General comment: The example provided in the target paragraph (paper by Simpson et al.) states that degradation depends on E3 localization, the example reported in the E3-features paragraph explains how AURKA is differently complexed during the cell cycle and therefore susceptible to degradation by different E3 ligases, it is more suitable for the first paragraph in my opinion. The E3 paragraph concludes with a dissertation on ubiquitin chains, while the ubiquitin paragraph mostly focuses on DUBs and Proteasome localisation.

We agree with this reviewer that the organisation of material in the review was challenging, since many of the parameters we describe could be considered from either perspective (target or Ub machinery). Our rationale for the specific examples listed are that – in the Simpson et al. paper the design of the experiments is focused on what happens to target in different compartments but since they do not compare the same substrate across different compartments, they can't draw definite conclusions about compartmentalised E3 activity. In the Liu et al. 2022 paper on the other hand – this paper does not conclude that AURKA is differentially degraded by different PROTACs because AURKA is complexed with different binding partners (although we would agree with that statement), but because the different harnessed E3s are differentially expressed in the cell cycle. We think this puts the paper clearly in the 'machinery' section.

I would suggest the authors reconsider the choice of the paragraphs' naming and their contents to make the text more consistent with the paragraphs and to make the whole paper more easily readable.

We have rationalised the choice of paragraph names (**Target-relevant parameters for PROTAC efficacy, UPS machinery influencing PROTAC efficacy**) to try and provide a more consistent narrative, as suggested by the reviewer.